# Bio-Stimulated Adsorption of Cr(VI) from Aqueous Solution by Groundnut Shell Activated Carbon@Al Embedded Material

**Dhilleswara Rao Vaddi** [1], **Thirumala Rao Gurugubelli** [2], **Ravindranadh Koutavarapu** [3,*], **Dong-Yeon Lee** [3,*] **and Jaesool Shim** [4,*]

[1]   Chemistry Division, Department of Basic Sciences and Humanities, GMR Institute of Technology, GMR Nagar, Rajam 532127, Andhra Pradesh, India; dhilleswararao.v@gmrit.edu.in
[2]   Physics Division, Department of Basic Sciences and Humanities, GMR Institute of Technology, GMR Nagar, Rajam 532127, Andhra Pradesh, India; thirumalaphy@gmail.com
[3]   Department of Robotics and Intelligent Machine Engineering, College of Mechanical and IT Engineering, Yeungnam University, Gyeongsan 712-749, Korea
[4]   School of Mechanical Engineering, Yeungnam University, Gyeongsan 712-749, Korea
[*]   Correspondence: ravindra_physicist@ynu.ac.kr (R.K.); dylee@ynu.ac.kr (D.-Y.L.); jshim@ynu.ac.kr (J.S.)

**Abstract:** In this study, a low-cost bioadsorbent aluminum metal blended with groundnut shell activated carbon material (Al-GNSC) was used for Cr(VI) adsorption from aqueous solutions. Al-GNSC was prepared and characterized using Fourier transform infrared spectrometer (FT-IR), scanning electron microscopic (SEM) and X-ray diffraction (XRD) to determine its surface morphology. Batch studies were performed and the optimum conditions for maximum Cr(VI) removal (of 94.2%) were found at pH 4.0, initial concentration 100 mg/L, adsorbent dosage 8 g/L of Cr(VI) solution, and time of contact 50 min. Moreover, the Langmuir isotherm model (maximum adsorption capacity of 13.458 mg/g) was the best fit and favored the mono-layered Cr(VI) adsorption. The kinetic studies reveal that the pseudo-second-order model was the best fit and favored chemisorption as the rate-limiting step. The desorption study revealed that Cr(VI) leached with sodium hydroxide solution acted as a regenerating agent. It is proved that Al-GNSC removes the Cr(VI) content in groundwater samples. The methodology developed using the Al-GNSC adsorbent as an alternative for the adsorption of Cr(VI) ions is remarkably successful in this study.

**Keywords:** biosorption; aluminum metal; groundnut shell; activated carbon; catalyst; Cr(VI); adsorption

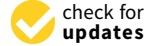



## 1. Introduction

The earth is recognized as a water planet because approximately 74% of the earth's surface is covered with water. High-quality water is an essential requirement for people and is considered a fundamental factor directly associated with economic development [1]. Water is normally accessible from two significant sources: surface and groundwater. Surface and subsurface water sources are limited and subject to change according to atmospheric and ecological conditions. Subsequently, groundwater is a significant water source for industrial, irrigation, and domestic needs. The water used for drinking purposes should be free from any poisonous components and minerals that might be perilous to living beings [2]. Groundwater quality is a function of both natural and human impacts. Groundwater is contaminated either directly or indirectly by improper disposal of garbage, dumping of industrial waste, use of fertilizers in agriculture, and natural processes [3].

Heavy metals, such as Cr, Ni, Cd, Cu, As, and Pb have drawn extensive attention owing to their high carcinogenicity, toxicity, and persistence [4]. One of the major groundwater pollutants is Cr(VI). Two oxidation forms of chromium exist in water: Cr(VI) and Cr(III). The occurrence of Cr(VI) in water is generally because of the dissolution of several Cr(VI) minerals from metal cleaning, leather, metallurgical, wood preservative, and paint

industries [5]. Hexavalent chromium is highly dangerous because of its carcinogenic character and is almost 100 times more toxic than the trivalent form [6]. A high concentration of Cr(VI) causes respiratory diseases, lung carcinomas, and allergies [7]. Chromium exists in various anionic forms in solution based on the pH of the solution. The desirable limit of chromium in potable water according to World Health Organization-2017 and United States-Environmental Protection Agency (US-EPA) is 0.05 mg/L and 0.1 mg/L, respectively [8]. Hence, innovative advancements in the decrease of Cr(VI) levels in groundwater are urgently needed.

Various strategies have been addressed for the expulsion of Cr(VI) from contaminated water and procedures, such as precipitation, electro-dialysis, electro de-fluoridation, membrane, filtration, ion-exchange, oxidation, coagulation and reverse osmosis, etc., are utilized [9–11]. These techniques have a few disadvantages, such as a particularly high amount of sludge disposal, the development of large-sized particles, blockage of membranes by metal hydroxide formation, and high operation costs. Hence, adsorption is considered an economical method because of its simple design, and a wide assortment of efficient adsorbents has been accounted for [12,13]. Activated carbon is a remarkable adsorbent of pollutants in water [14]. It is a black solid (homogeneous or heterogeneous surface), either pelletized or microcrystalline, with a large surface area and high porosity [15]. Commercially available activated carbon is extremely expensive; hence, activated carbon prepared from low cost materials (such as groundnut shell, rice husk, sawdust, bark and stems of plants) could profoundly reduce the expense of the Cr(VI) adsorption process.

Various materials were used to remove the Cr(VI) ions, such as Gonzalez et al. studied Cr(VI) adsorption by coconut coir as an adsorbent from an aqueous solution. The maximum removal of Cr(VI) took place under optimum conditions of pH 2.0, adsorbent dosage 6.5 g/L, initial concentration 200 mg/L and contact period 20 min. The maximum adsorption capacity obtained was 6.25 mg/g [16]. Rai et al. studied Cr(VI) adsorption by activated carbon prepared from mango kernel as an adsorbent from aqueous solution. The maximum removal of Cr(VI) took place under optimum conditions of pH 2.0, an adsorbent dosage of 10 g/L, an initial concentration of 60 mg/L and a contact period of 150 min. The maximum adsorption capacity obtained was 7.8 mg/g [17]. Dubey et al. studied Cr(VI) adsorption by activated carbon prepared from groundnut husk activated carbon as an adsorbent and groundnut husk activated carbon impregnated with silver from an aqueous solution. The maximum removal of Cr(VI) took place under optimum conditions of pH 3.0, an adsorbent dosage of 5 g/L, an initial concentration of 10 mg/L and a contact period of 300 min. The maximum adsorption capacity obtained is 7.01 mg/g by using groundnut husk activated carbon and under the same optimum conditions, 11.39 mg/g of adsorption capacity is obtained by using groundnut husk activated carbon impregnated by silver particles [18].

In order to increase the adsorption capacity, recently, surface modification of activated carbon-based materials with suitable chemicals has been successfully investigated to increase the adsorption efficiency, suggesting that the alteration of activated carbon can effectively reduce the Cr(VI) content [19]. Negative ions (such as fluoride, chromate, and arsenate) have a large affinity towards metal ions, such as $La^{3+}$, $Fe^{3+}$, and $Al^{3+}$; therefore, activated carbon mixed with metal ions shows excellent potential for the removal of chromate ions [20]. As indicated by the reality of the issue and lack of an appropriate economical material for the adsorption of Cr(VI) ions, we studied the preparation of low-cost adsorbents using groundnut shells as bioadsorbents because financial suitability is a prerequisite for the absorption of chromium from water in rural areas. Groundnut shell is predominantly available in many regions as a less biodegradable bio-waste, contains surface groups, and does not discharge soluble contaminants into the water, making it an appropriate selection for this present study. Moreover, utilizing low-expense aluminum metal ions as a binding site for chromate ion interactions may increase the sorption of Cr(VI) ions. Since the coupling of metal ions with carbonaceous adsorbent can provide a large number of active sites which can interact chemically with the adsorbates thereby enhancing

chemisorption. The present study examines the efficiency of aluminum metal blended with groundnut shell activated carbon material (Al-GNSC) as a bioadsorbent for the adsorption of Cr(VI) from aqueous solutions. The adsorption of Cr(VI) in groundwater samples is also examined and the adsorption capacity of the prepared adsorbent is compared with the existing available adsorbents.

## 2. Results and Discussion

### 2.1. Fourier Transform Infrared Spectrometer(FT-IR) Analysis

FT-IR was used to determine the structural information, such as the presence of different functional groups on the adsorbent surface. The FT-IR spectra of the Al-GNSC adsorbent before and after the adsorption of Cr(VI) are shown in Figure 1, and the corresponding data are presented in Table 1. The band spectrum of Al-GNSC before adsorption (Figure 1a) had a vibrational band at 3417 cm$^{-1}$, corresponding to the –OH stretching band in hydroxyl groups [21,22] and was shifted to 3421 cm$^{-1}$ after Cr(VI) adsorption. The bands at 2920 cm$^{-1}$ and 2849 cm$^{-1}$ before adsorption represent the presence of the –CH$_2$ stretching band and were shifted to 2929 cm$^{-1}$ and 2852 cm$^{-1}$, respectively. The stretching band at 1708 cm$^{-1}$ represents the C=O group in carboxylic acid and was shifted to 1713 cm$^{-1}$. The stretching band at 1615 cm$^{-1}$ represents the C=C group. The peaks at 1262 cm$^{-1}$ and 1026 cm$^{-1}$ indicated the presence of –C–O in alcoholic or carboxylic acid and were shifted to 1266 cm$^{-1}$ and 1029 cm$^{-1}$, respectively. There was a marginal shift in the vibrational bands after treatment of the adsorbent with Cr(VI) solution (Figure 1b), indicating the occurrence of Cr(VI) ion sorption. Furthermore, no vibrational band diminishing was observed during the sorption, indicating that Al-GNSC is an excellent adsorbent for the treatment of Cr(VI) wastewater.

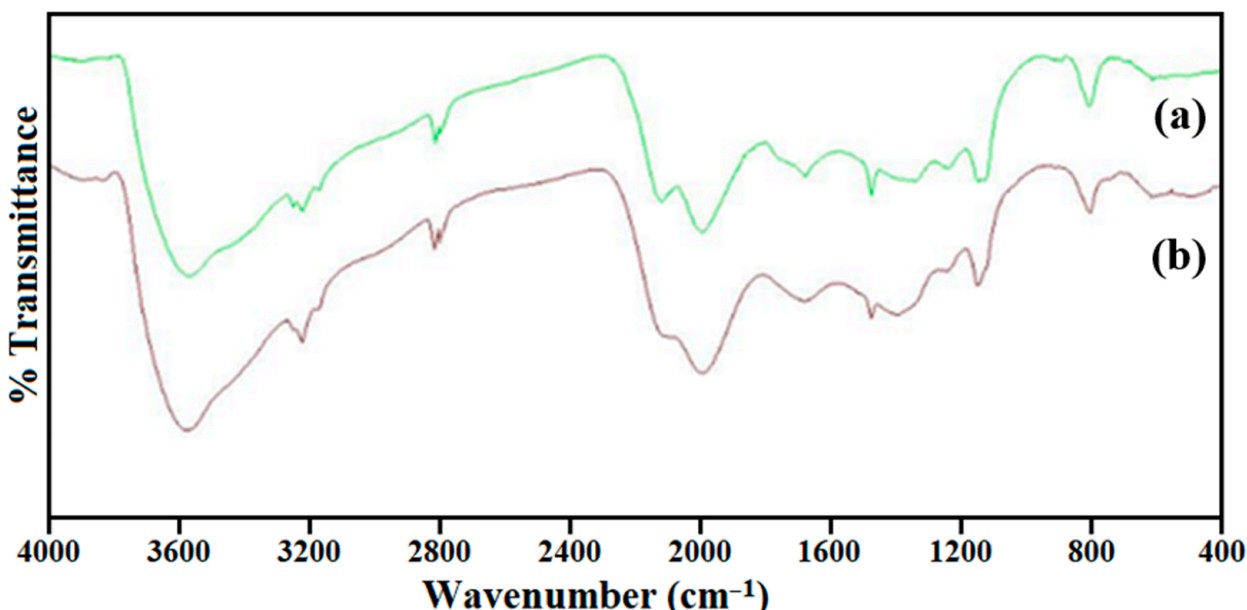

**Figure 1.** FT-IR spectra of Al-GNSC: (**a**) before, and (**b**) after Cr(VI) adsorption.

### 2.2. Morphology Analysis

The surface morphology of the Al-GNSC was examined using scanning electron microscopic (SEM) analysis and energy dispersive x-ray analysis (EDX). The SEM images of Al-GNSC before and after the adsorption of Cr(VI) are shown in Figure 2. The adsorbent before treatment with Cr(VI) solution (Figure 2a) had small-sized particles of 15–122 nm, and a few larger-sized particles with an irregular shape and porous nature, which provided efficient active sites. The adsorbent (Al-GNSC) after treatment (Figure 2b) also represented almost similar size and shape but with less porosity. The small changes in morphology may be attributed to the sorption and ion exchange of chromate ions with inorganic layers,

such as Al(OH)$_3$ and AlCl$_3$ because the activated carbon material was chemically modified by the aluminum metal ions [23].

**Table 1.** FT-IR data of Al-GNSC before and after Cr(VI) adsorption.

| Before Al-GNSC | After Al-GNSC | Bond Stretching Values |
|:---:|:---:|:---:|
| 3417 | 3421 | -OH stretching vibration |
| 2920 | 2929 | CH$_2$ anti symmetric stretching vibration |
| 2849 | 2852 | CH$_2$ symmetric stretching vibration |
| 2360 | 2360 | C=O stretching |
| 1708 | 1713 | C=O stretching |
| 1613 | 1621 | Aromatic C=C stretching vibration |
| 1405 | 1398 | CH$_2$ bending vibration |
| 1262 | 1266 | C-O stretching vibration in alcohol |
| 1169 | 1198 | C-O stretching vibration of tertiary alcohol |
| 1026 | 1029 | C-O stretching vibration in alcohol |
| 660,796 | 660,796 | C=C bending vibration |
| 459 | 487 | Al-O frequency vibration |

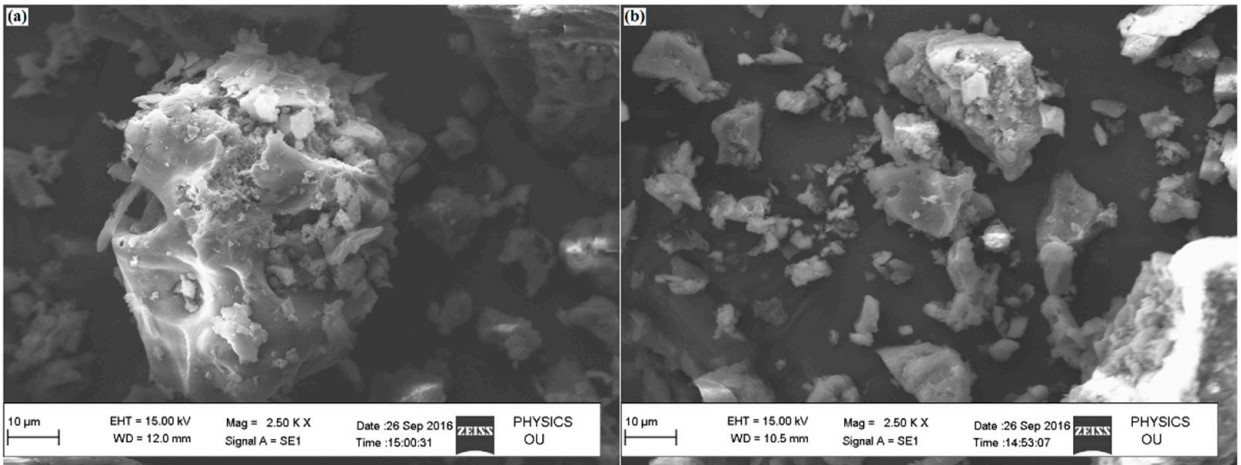

**Figure 2.** SEM Images of Al-GNSC: (**a**) before, and (**b**) after Cr(VI) adsorption.

The EDX spectra of the adsorbent provided data on the elemental compositional changes of the Al-GNSC surface before and after Cr(VI) adsorption. The appearance of chromium in the EDX spectra of Al-GNSC after Cr(VI) adsorption confirms the adsorption, as the Cr(VI) peak was not observed in the raw Al-GNSC adsorbent, as shown in Figure 3.

*2.3. X-ray Diffraction (XRD) Studies*

Powder XRD was done on Panalytical Xpert Pro diffractometer with CuKα radiation (1.5406 A). The powder XRD data of raw Al-GNSC adsorbent and after Cr(VI) adsorption is shown in Figure 4. The XRD spectra demonstrate that a significant portion of the Al-GNSC material is amorphous in nature and some sharp peaks appear which indicates the presence of some crystalline part. The XRD peaks noticed at 2θ values of 51.84 and 55.02 which might represent Al(OH)$_3$, 40.73 and 38.16 represents Al metal [20], and 30.84 and 24.88 are related to α-Al$_2$O$_3$ and AlCl$_3$ crystalline parts present in the amorphous material. Thus, the Al-GNSC is somewhat covered with the crystalline regions of Al(OH)$_3$, α-Al$_2$O$_3$ and AlCl$_3$ and a major part of the material is an amorphous region. The amorphous region provides more active sites for Cr(VI) adsorption and the crystalline parts enable the exchange of chromate ions, with complete adsorption of chromium ion by the adsorbent. The crystal size of the material is measured as 17.3 nm.

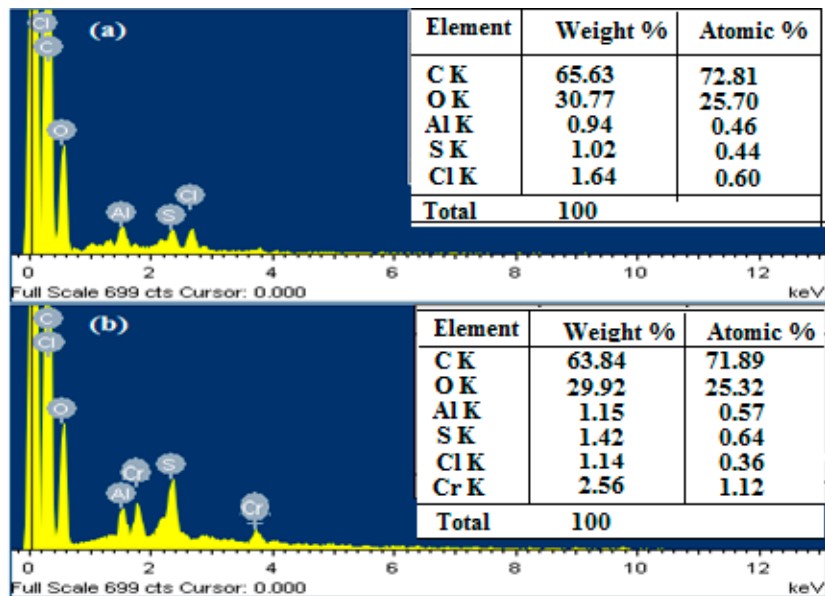

**Figure 3.** EDX Images of Al-GNSC: (**a**) before, and (**b**) after Cr(VI) adsorption.

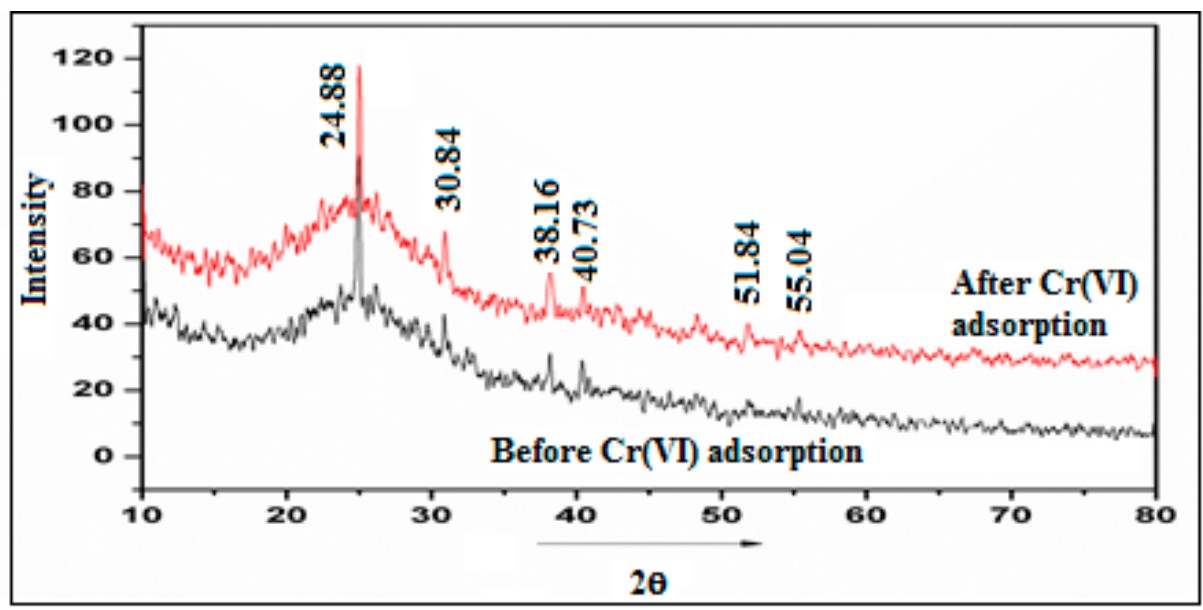

**Figure 4.** XRD images of Al-GNSC before, and after Cr(VI) adsorption.

*2.4. Adsorption Studies*

2.4.1. Influence of Initial pH for Cr(VI) Removal

pH is an important factor that affects Cr(VI) adsorption because it determines the surface charge of the adsorbent (Al-GNSC). The zero-point charge (ZPC) of the Al-GNSC adsorbent was measured using a standard technique to identify the charge on the adsorbent surface [24]. A diagram plot between the initial and final pH provided a curve where the $pH_{ZPC}$ for Al-GNSC was measured as the point at which the change in pH was zero. The result demonstrated that the $pH_{ZPC}$ of Al-GNSC was 6.81 which means the Al-GNSC surface was positively charged below pH 6.81 and negatively charged above pH 6.81 [25]. Batch adsorption studies were performed in the pH range of 3.0 to 10.0 using a predetermined Al-GNSC adsorbent dosage (8 g/L of 100 mg/L Cr(VI)) and a contact time of 50 min. The percentage removal of Cr(VI) and Cr(VI) uptake on the adsorbent surface at different pH values is presented in Figure 5a. Chromium exhibited different pH-dependent equilibria

in solution. If the solution pH varied, there was a change in equilibrium. $HCrO_4^-$ and $Cr_2O_7^{2-}$ ions were in equilibrium at a pH range of 2 to 6; and $Cr_3O_{10}^{2-}$ and $Cr_4O_{13}^{2-}$ were available at lower pH values. The maximum adsorption of Cr(VI) occurred at a pH of 4.0, which was below the $pH_{ZPC}$ of Al-GNSC. At lower pH, the adsorbent surface was positively charged; thus, there was a high coulombic attraction between the adsorbent surfaces and chromate ions. The adsorption of Cr(VI) was less at pH $\geq$ 8 because of an increase in the repulsion between the hydroxyl ions and chromate ions.

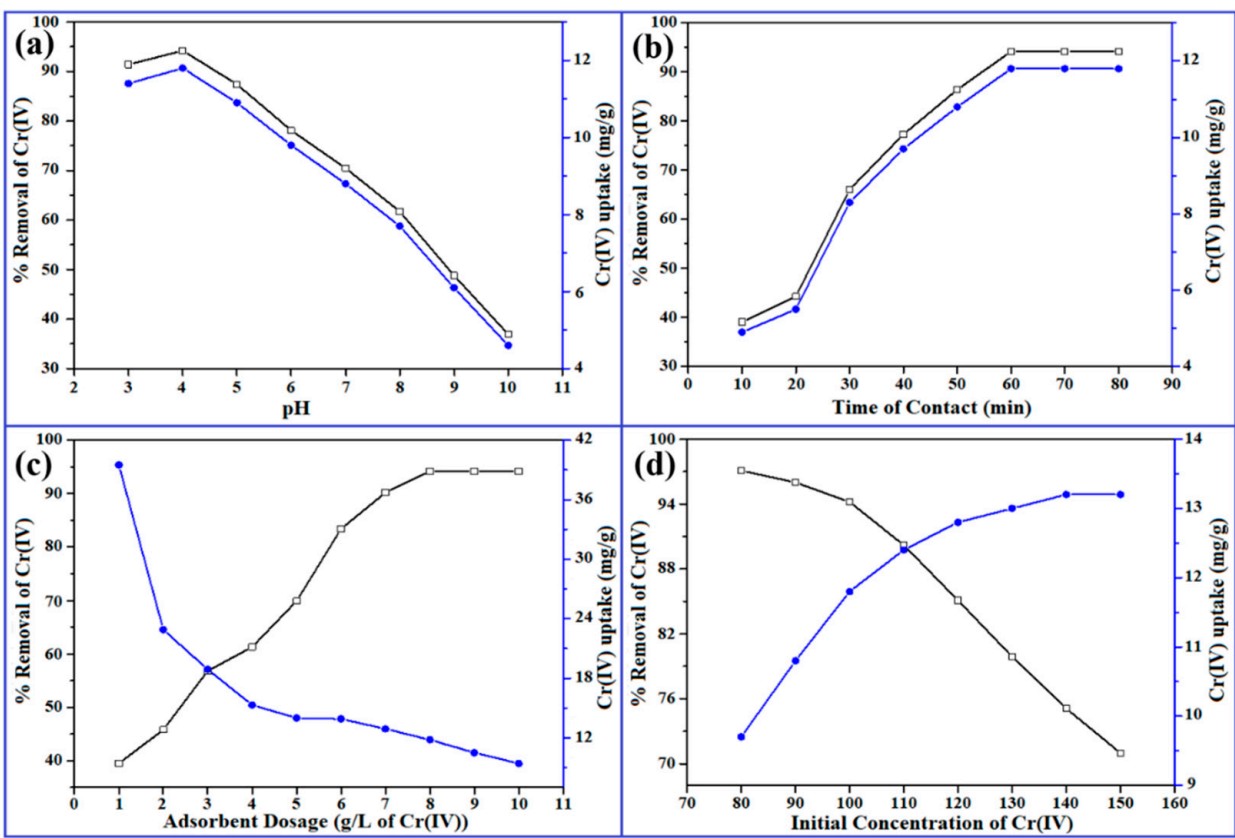

**Figure 5.** Parameters affecting percentage removal of Cr(VI) and Cr(VI) uptake (mg/g) using Al-GNSC, the influence of (**a**) pH, (**b**) contact time, (**c**) Al-GNSC dosage, and (**d**) initial Cr(VI) concentration.

### 2.4.2. Influence of Contact Time for the Removal of Cr(VI)

The contact time between the adsorbent (Al-GNSC) and the Cr(VI) solution also plays a key role in determining the equilibrium time for the maximum removal of Cr(VI). The experimental work was conducted with varied time periods between 10 and 80 min, a predetermined adsorbent dosage (8 g/L of 100 mg/L Cr(VI) solution), and a pH of 4.0. The results (Figure 5b) demonstrated that the Cr(VI) adsorption as well as Cr(VI) uptake increased gradually up to 50 min, after which there was no significant change. This may be due to the high initial availability of binding sites for adsorption. After attaining equilibrium, the expulsion of Cr(VI) may have become insignificant because of the unavailability of adsorbing sites [26].

### 2.4.3. Influence of Adsorbent Dosage for the Removal of Cr(VI)

The experimental work was conducted by varying the adsorbent dosage from 1 g/L to 10 g/L by maintaining an optimum pH of 4.0, a contact period of 50 min, and a concentration of 100 mg/L Cr(VI). As the adsorbent (Al-GNSC) dosage increased from 1 g/L to 8 g/L, the adsorption of Cr(VI) gradually increased as the number of binding sites increased with increasing adsorbent dosage (Figure 5c). After attaining equilibrium, there

was no significant increase as the number of Cr(VI) ions to adsorb in the solution remained constant. However, the Cr(VI) adsorbed on the surface 'Qe' decreased from 39.5 mg/g to 9.42 mg/g with increased adsorbent dosage, as the number of Cr(VI) ions in the solution available for adsorption on the particular adsorbent surface was constant.

### 2.4.4. Influence of Concentration of Cr(VI) for the Removal of Cr(VI) Ion

The experimental work was performed by varying the Cr(VI) concentrations from 80 mg/L to 150 mg/L under optimum conditions of adsorbent (Al-GNSC) dosage of 8 g/L of Cr(VI) solution, pH 4.0, and contact time of 50 min. The removal percentage of Cr(VI) (Figure 5d) decreased with increasing concentration, which was mainly due to the saturation of binding sites because the adsorbent dosage (number of binding sites) was fixed. However, the Cr(VI) uptake 'Qe' increased with increasing concentrations owing to more Cr(VI) ions being available to interact with the adsorbent surface [27].

### 2.5. Adsorption Isotherm Studies

In this study, two important isotherm models, the Freundlich and Langmuir models, were examined to identify the best fit model [21]. The adsorption isotherm is helpful for representing the relationship between the equilibrium amounts of Cr(VI) on the adsorbent surface and plays a vital role in identifying the adsorption limit. The Freundlich model assumes that adsorption takes place on heterogeneous sites of the adsorbent material and has a multi-layered uptake, whereas the Langmuir isotherm assumes that adsorption takes place on homogeneous sites. A graph plot between '$lnC_e$' vs. '$lnQ_e$' (Figure 6a) corresponded to a straight line with an intercept lnK and slope (1/n) in the Freundlich model, and '$1/Q_e$' vs. '$1/C_e$' corresponded to a straight line in the Langmuir model (Figure 6b) using Al-GNSC with different concentrations of Cr(VI) ion.

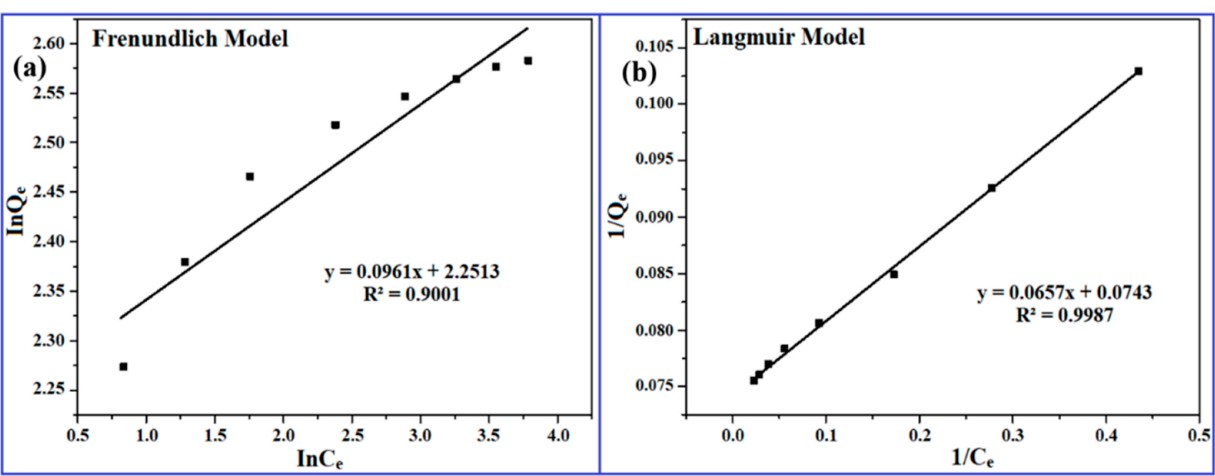

**Figure 6.** (**a**) Freundlich and (**b**) Langmuir adsorption isotherm curves using Al-GNSC adsorbent.

The linear forms and adsorption isotherm data for the two models are presented in Table 2. The $R^2$ (0.996) in the Langmuir model was higher than that in the Freundlich model (0.909), indicating that adsorption takes place on the homogenous adsorbent surface and that the process favors uni-layered adsorption. The important characteristic feature of Langmuir is the separation factor '$R_L$'. $R_L = \frac{1}{1+bC_i}$, where "$C_i$" is the initial concentration of Cr(VI) (mg/L) and "$b$" is the Langmuir constant. Based on the "$R_L$" experimental values, it can be demonstrated whether the adsorption is favorable ($0 < R_L < 1$), unfavorable ($R_L > 1$), or irreversible ($R_L = 0$). The obtained '$R_L$' value of 0.024, represented that adsorption was favorable.

**Table 2.** Freundlich and Langmuir adsorption isotherm data.

| Adsorption Models | Linear Forms | Graph Drawn between | $R^2$ | Parameters | Obtained Value |
|---|---|---|---|---|---|
| Freundlich Model | $lnQ_e = lnK_f + \frac{1}{n_f}lnC_e$ | $lnC_e$ versus $lnQ_e$ | 0.9001 | $K_f$ | 9.487 |
| | | | | $1/n_f$ | 0.0961 |
| Langmuir Model | $\frac{1}{Q_e} = \frac{1}{Q_m b C_e} + \frac{1}{Q_m}$ | $1/C_e$ versus $1/Q_e$ | 0.9987 | $b$ | 0.1131 |
| | | | | $Q_m$ | 13.458 |

In this, '$R^2$' indicates regression constant, $1/n_f$' indicates the quantity of adsorption intensity, and '$K_f$' indicates adsorption capability. '$b$' is constant represents the affinity of binding sites (L/mg), '$R_L$' is dimensionless equilibrium parameter.

### 2.6. Kinetic Adsorption Study

The Kinetic model is generally utilized to clarify the adsorption mechanism. The models studied in this present work are recorded in Table 3 alongside their graphical portrayals(Figure 7), in particular, pseudo-first-order( mainly associated with physisorption) and pseudo-second-order(mainly associated with chemisorption) [28] models. The results demonstrated that the correlation coefficient value ($R^2 = 0.9941$) closer to one represents the pseudo-second-order model best fit and chemisorption is the rate-limiting step of the adsorption process utilizing Al-GNSC.

**Table 3.** Pseudo first order and second order data.

| Adsorption Kinetic Models | Linear Forms | Graph Drawn between | $R^2$ | Parameters | Obtained Value |
|---|---|---|---|---|---|
| Pseudo-first-order | $ln(Q_e - Q_t) = lnQ_e - K_1 t$ | $t$ verses $ln(Q_e - Q_t)$ | 0.9901 | $K_1$ | 0.0479 |
| | | | | $Q_e$ | 12.1 |
| Pseudo-second-order | $\frac{t}{Q_t} = \frac{1}{K_2 Q_e^2} + \frac{1}{Q_e}t$ | $t$ verses $\frac{t}{Q_t}$ | 0.9941 | $K_2$ | 0.002 |
| | | | | $Q_e$ | 15.45 |

In this, '$R^2$' indicates regression constant, '$K_1$' (min$^{-1}$) is the pseudo-first-order rate constant. '$K_2$' (g·(mg/min)$^{-1}$) is the pseudo-second-order rate constant. '$Q_e$' and '$Q_t$' indicates Cr(VI) uptake at equilibrium and at time "$t$".

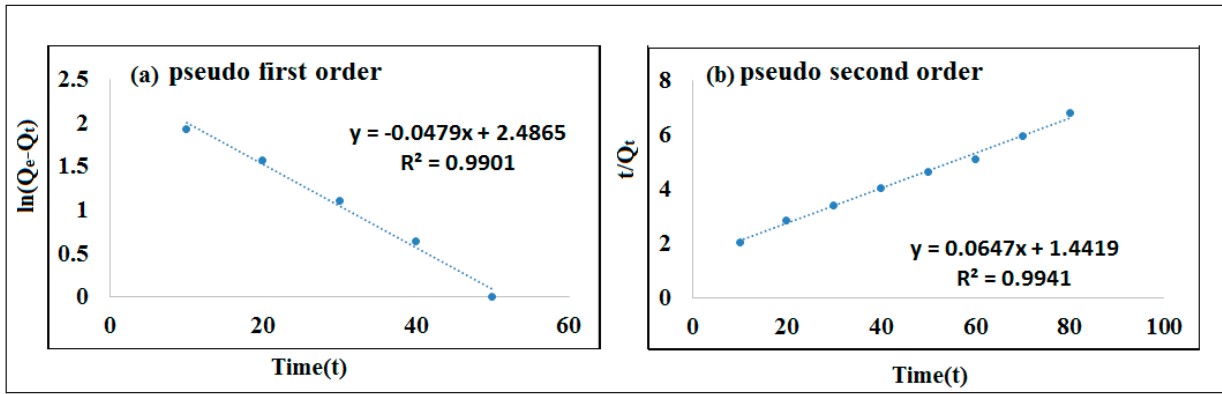

**Figure 7.** (**a**) pseudo first order and (**b**) pseudo second order curves using Al-GNSC adsorbent.

### 2.7. Regeneration of Adsorbent

The adsorption method is economical if the adsorbent is reused. Reuse of Al-GNSC from Cr(VI)-loaded material is studied for sorption and desorption cycles using sodium hydroxide as a regenerating agent. Four different desorption agents, such as tap water, 0.1 M HCl, 0.1 M H$_2$SO$_4$ and 0.1 M NaOH are used to remove the adsorbed chromium ions from the Al-GNSC adsorbent. Cr(VI) desorption studies were performed using 10 g of Cr(VI)-loaded adsorbent with 100mL desorption agents in 250 mL conical flasks under

agitation for 180 min at 180 rpm. From these various desorption agents, it is identified that 0.1 M NaOH was more effective [29,30]. Hence, sodium hydroxide solution is considered a desorption agent. After reusing of regenerated adsorbent for Cr(VI) removal, the Cr(VI) loaded adsorbent was again regenerated by the same procedure demonstrated previously for consecutive cycles. Numerous regenerations and the following use for the removal of Cr(VI) are completed and the obtained results are shown in Figure 8. It is decided from the graph that there is a reduction in the percentage removal of Cr(VI) with an increase in the regeneration cycle, the removal percentage was still above 70%, even after three cycles. Therefore, Al-GNSC has a more re-usage potential for the Cr(VI) removal.

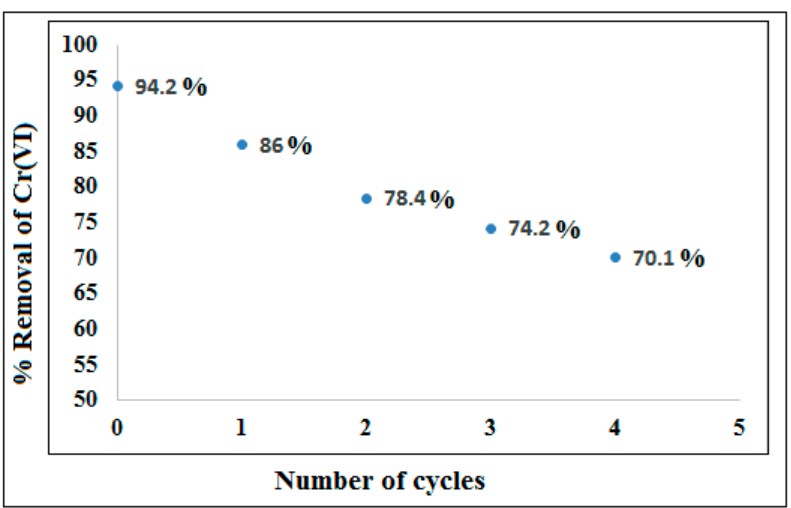

**Figure 8.** % Removal of Cr(VI) from the prepared Al-GNSC adsorbent and regenerated adsorbent of several cycles.

*2.8. Cost Estimation*

Economic viability is a prerequisite for the decontamination of water in rural areas. Groundnut shells are widely available as a bio-waste. Cost estimation was performed for the prepared adsorbent (Al-GNSC), which offers a good option for Cr(VI) adsorption. The groundnut shells were collected free of cost. The cost of the drying process and chemicals utilized led to an approximate cost of 400 INR per kilogram of prepared adsorbent, which was lower than that of the commercially available activated carbon generally utilized in the Cr(VI) treatment.

*2.9. Feasible Mechanism for the Adsorption of Cr(VI)*

Chromium occurs in various oxidation forms based on the pH of the solution [31]. The equilibriums between various ionic types of chromium are as follows:

$$H_2CrO_4 \Leftrightarrow H^+ + HCrO_4 \tag{1}$$

$$2HCrO_4{}^- \Leftrightarrow Cr_2O_7{}^{2-} + H_2O \tag{2}$$

$$HCrO^{2-} \Leftrightarrow H^+ + CrO^{2-} \tag{3}$$

At low pH, the anionic form of chromium ($HCrO_4{}^-$) binds to the positively charged active binding sites of the adsorbent via electrostatic forces [32,33]. The adsorption capacity of original groundnut shell activated carbon before blending with aluminum is 7.4 mg/g; however, the adsorption capacity of aluminum mixed groundnut shell activated carbon is 13.458 mg/g. The FT-IR studies demonstrated the presence of various polar functional groups, such as hydroxyl, carboxylic and phenolic groups in the Al-GNSC material. The FTIR peak of hydroxyl (–OH) at 3417 cm$^{-1}$ before adsorption of Cr(VI) is shifted to 3421 cm$^{-1}$. The stretching band at 1708cm$^{-1}$ belongs to the C=O group in the metal carboxylate and is shifted to 1713 cm$^{-1}$. The peaks at 1262 cm$^{-1}$ and 1026 cm$^{-1}$

represent the presence of –C–O in alcoholic or carboxylic acid and these peaks are shifted to 1266 cm$^{-1}$ and 1029 cm$^{-1}$. These peaks shift demonstrated the strong interaction of the adsorbent with the Cr(VI) ions. Furthermore, the kinetic study supported that adsorption takes place through chemisorption since the pseudo-second-order model is the best fit for Cr(VI) adsorption. Hence, the nature of the Al-GNSC surface, various functional groups, and their contact with adsorbing chromate ions govern the feasible mechanism of adsorption. The pH$_{ZPC}$ of Al-GNSC is significant as it perceives the ionic state of Al-GNSC. In this study, maximum adsorption occurred at a pH (4.0) lower than pH$_{ZPC}$. Therefore, the active binding sites on the adsorbent surface were positively charged [34]. Based on these results, the Al-GNSC/Cr(VI) reaction might be indicated in two feasible ways as presented in Figure 9.

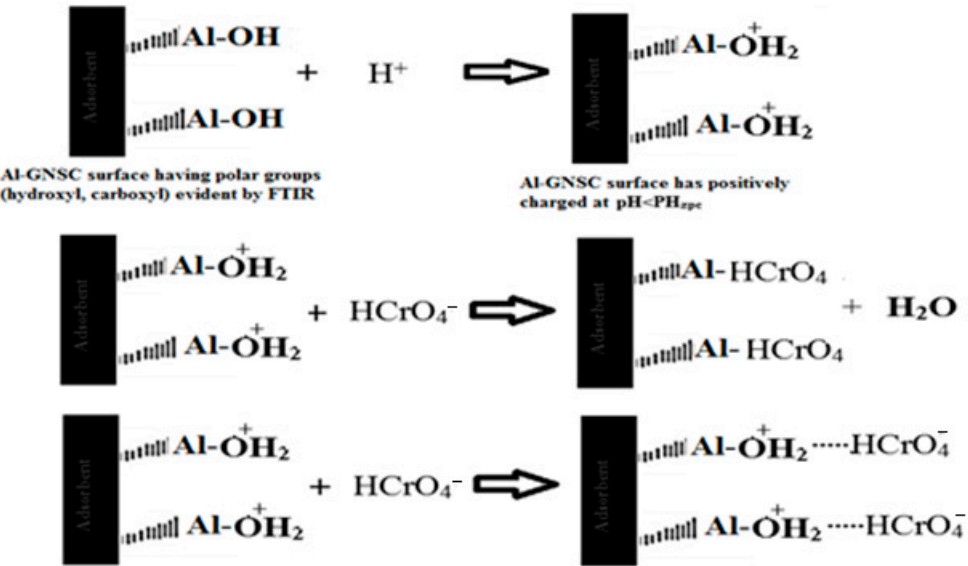

**Figure 9.** Possible mechanism of the adsorption of chromium ions by Al-GNSC.

*2.10. Comparison Studies of Al-GNSC with Other Available Adsorbents*

A comparative report has been considered for the adsorption limits of different adsorbent materials and the current adsorbent Al-GNSC. The immediate comparison of AL-GNSC with different adsorbents is not simple considering the different working conditions. An effort was made (Table 4) for the assessment of certain adsorbents. In the context of the obtained results, the adsorption ability of Al-GNSC for Cr(VI) was more significant than that of other accessible adsorbents.

*2.11. Application of Al-GNSC for the Removal of Cr(VI) in Groundwater Samples*

The flexibility of the adsorbent (Al-GNSC) in this work for removing Cr(VI) has been attempted with groundwater samples collected from Gara Mandal, Srikakulam District of Andhra Pradesh, India. According to the WHO [8], the permissible limit of Cr(VI) in drinking water is 0.05 mg L$^{-1}$. Chromium concentrations in the groundwater samples exceeded the permissible limits (i.e., 0.05 mg/L). Therefore, in the present work, the removal of Cr(VI) from groundwater samples is carried out at optimum conditions of adsorbent dosage (8 g/L), pH (4.0) and time of contact (60 min) using an Al-GNSC adsorbent to reduce the Cr(VI) content to below the allowable limit. The obtained data belonging to concentrations of Cr(VI) ions in the groundwater samples before as well as after being treated with Al-GNSC are presented in Table 5. It is proved that Al-GNSC removes the Cr(VI) content in groundwater samples. It is understood that the methodology developed using the Al-GNSC adsorbent for the adsorption of Cr(VI) ions in this study work is remarkably successful.

Table 4. Comparison studies of Al-GNSC with other existing adsorbents for the adsorption of Cr(VI).

| Adsorbent Material | Initial Cr(VI) Conc. (ppm) | pH | Contact Time (min) | Adsorbent Dosage (g L$^{-1}$) | Maximum Adsorption Capability (mg g$^{-1}$) | References |
|---|---|---|---|---|---|---|
| Activated carbon (AC) prepared from coconut tree sawdust | 10 | 3.0 | 180 | 0.2 | 3.46 | [35] |
| Raw coconut fiber | 250 | 1.0 | 270 | 10 | 18.60 | [36] |
| Sugarcane bagasse | 100 | 2.0 | 90 | 10 | 1.76 | [37] |
| Canadian peat Coconut fiber | 50 | 2.0 | 4320 | 25 | 4.61 4.71 | [38] |
| peanut shell (P. Shell), sawdust (S. Dust) and *Cassia fistula* leaves (C.F. Leaves). | 40 | 2.0 | 360 | 5 | 4.32 3.66 & 4.48 | [39] |
| Al-GNSC | 100 | 4.0 | 60 | 8 | 13.458 | Present Work |

Table 5. The concentration of Cr(VI) in groundwater samples before and after adsorption using Al-GNSC adsorbent.

| Groundwater Sample. No. | Latitude (°N) | Longitude (°E) | Water Sample Collected in the Village | Before Adsorption of Cr(VI) (mg/L) | After Adsorption of Cr(VI) (mg/L) |
|---|---|---|---|---|---|
| 1 | 18.3063 | 84.0437 | Raghavapuram | 0.057 | Undetectable limit |
| 2 | 18.3220 | 84.0145 | Reddy Peta | 0.061 | Undetectable limit |
| 3 | 18.3150 | 83.8920 | Gonti–Entrance Road point | 0.058 | Undetectable limit |
| 4 | 18.2752 | 83.9633 | Korni (Sri sainaveedi) | 0.093 | Undetectable limit |
| 5 | 18.3122 | 84.0799 | Peddatulugu-Gollaveedi | 0.056 | Undetectable limit |

## 3. Experimental Procedure

### 3.1. Preparation of Aluminum Metal Blended Groundnut Shell Activated Carbon Adsorbent

Groundnut shells were collected from a local area as a bio-waste. They were dried in sunlight for approximately 10 days. The dried shells were ground into powder. Concentrated sulfuric acid was added dropwise to the groundnut shell powder in a weight ratio of 1:1.8 (groundnut shell powder: conc. $H_2SO_4$) [18,19]. The contents were then heated at $300 \pm 5$ °C in a muffle furnace for three hours and cooled to room temperature. The resultant material (groundnut shell activated carbon) was washed with distilled water several times until it was free of acidic molecules and dried at $120 \pm 5$ °C in an oven. Finally, aluminum powder was blended with the activated carbon in a weight ratio of 1:10, and concentrated hydrochloric acid was added dropwise. The obtained material was washed with distilled water several times until it was free from acidic molecules and dried in an oven. The dried substance was ground into a required particle size of <90 μm.

### 3.2. Chemicals and Instruments

A 1000 mg/L stock Cr(VI) solution was prepared using potassium dichromate. The working concentrations of Cr(VI) were prepared by proper dilution of the stock solution. Analytic reagent grade aluminum metal powder was used as a blending agent. The concentration of chromium ions was determined using an Ultraviolet-Visible double beam

spectrophotometer (Systronics, AU 2701, Gujarat, India), and a pH meter (Systronics 335, Gujarat, India) was used to determine the pH of the solution. Hydrochloric acid(HCl) and sodium hydroxide (NaOH) solutions (0.05 M) were used to determine the pH of the solution. The Fourier Transform-Infrared spectrum (Perkin Elmer, model- BX FT-IR, Waltham, MA, USA) was recorded from 4000 to 400 cm$^{-1}$. Scanning Electron Microscopic images were recorded using a SEM-Model Philips XL30 (Leuven, Belgium) to determine the surface morphology and elemental composition of the Al-GNSC.

*3.3. Batch Adsorption Studies*

Batch adsorption studies were performed by using 50 mL of 50 mg/L Cr(VI) solution with a fixed quantity of Al-GNSC adsorbent and a predetermined pH of the solution. The experimental solution was agitated in a mechanical orbital shaker at 200 rpm for a fixed period. After a particular time, the solution was filtered using Whatman filter paper-40, and the remaining Cr(VI) ions in the solution were measured. In each experiment, one of the parameters was varied, and the other parameters were fixed at a predetermined value. The parameters, such as solution pH (3.0–10.0), Al-GNSC adsorbent dosages (4–18 g/L), contact time (10–70 min), and initial Cr(VI) concentration (40–80 mg/L) were varied to obtain the optimum adsorption conditions for the maximum removal of Cr(VI) ions. The percentage removal of Cr(VI) and the amount of Cr(VI) adsorbed per gram of the Al-GNSC adsorbent at equilibrium '$Q_e$' (mg/g) was measured using the following equations:

$$\% \, R = \left[ \frac{C_i - C_e}{C_i} \right] \times 100 \tag{4}$$

$$Q_e = \left[ \frac{C_i - C_e}{m} \right] \times V \tag{5}$$

where '$C_i$' and '$C_e$' are the initial and equilibrium concentration of Cr(VI) in mg/L, '$m$' is the quantity of adsorbent (g), and '$V$' is the volume of Cr(VI) solution in L.

**4. Conclusions**

Maximum Cr(VI) adsorption (94.2%) took place under optimum conditions of pH 4.0, a contact time of 50 min, an adsorbent dosage of 8 g/L of Cr(VI) solution, and a Cr(VI) concentration of 100 mg/L. The optimum pH < pH$_{ZPC}$ indicated that the Al-GNSC adsorbent surface was positively charged, which in turn indicated that there was a high coulombic attraction between the adsorbent surfaces and Cr(VI) ions. FT-IR analysis confirmed the presence of different polar functional groups, such as hydroxyl and carboxylic groups, indicating that adsorption might occur through ion exchange with Cr(VI) ions. SEM examination confirmed that Al-GNSC has small particle agglomerates with a porous nature. EDX spectra confirmed the presence of a chromium peak after treatment of the adsorbent with chromium solution. The Cr(VI) equilibrium adsorption over the entire concentration range revealed that the Langmuir model is the best fit with maximum adsorption capacity (13.458 mg/g)and the adsorption is uni-layered. The kinetic adsorption studies reveal that the pseudo-second-order model was the best fit and chemisorption is the rate limiting step. XRD study reveals that Al-GNSC is moderately covered with crystalline parts. The amorphous nature of the adsorbent sample offers more active sites for Cr(VI) adsorption. Though the crystalline AlCl$_3$, Al (OH)$_3$ and, $\alpha$-Al$_2$O$_3$ particles also facilitate the exchange of chromium ions. Regeneration studies revealed that the adsorbent could be reused by various adsorption-desorption cycles by using 0.1M NaOH as a regenerating agent. It is proved that Al-GNSC removes the Cr(VI) content in groundwater samples. Hence, Al-GNSC is an extremely efficient adsorbent as compared with most of the available adsorbents as it is economically feasible and has the remarkable ability for Cr(VI) removal in wastewater treatment.

**Author Contributions:** Conceptualization, methodology, and writing—original draft preparation, D.R.V.; methodology and writing—original draft preparation, T.R.G.; validation, formal analysis, investigation, and review—data curation, supervision, and resources, R.K.; validation, formal analysis, investigation, and review—data curation, supervision, and resources, D.-Y.L.; supervision, editing, funding acquisition, J.S. All authors have read and agreed to the published version of the manuscript.

**Funding:** This research was supported by the National Research Foundation of Korea (NRF) funded by the Korean Government (2020R1A2C1012439), Republic of Korea. This work was supported by the Korea Institute for Advancement of Technology (KIAT) grant funded by the Korea Government (MOTIE) (No. P160500014), Republic of Korea.

**Acknowledgments:** One of the authors, Thirumala Rao Gurugubelli, wishes to express his gratitude to the GMRIT management for providing financial assistance through the SEED grant for the research.

**Conflicts of Interest:** The authors have no conflict of interest to declare that are relevant to the content of this article.

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
