# Peer review of "Bio-Stimulated Adsorption of Cr(VI) from Aqueous Solution by Groundnut Shell Activated Carbon@Al Embedded Material"

_catalysts, doi:10.3390/catal12030290_

Round 1
Reviewer 1 Report
My comments on this paper are:
- Some FT-IR peaks shift with adsorption of Cr, and some not. Please add the discussion which peak shift is related to Cr, the authors can combine these with the discussion of the mechanism.
- The porosity of the sample should be measured.
- It is better to show the SEM images using the same scale. The magnified images can be shown separately or as inset.
- In Figure 3, what is the origin of the increase concentration of Al and S?
- In the experiment, H2SO4 is added in the first step, what is the purpose? Will this bring pollution problem?
- How about the Cr concentration in the polluted ground water? Is the products suitable for this real condition?
Author Response
We appreciate the efforts of the reviewers for their detailed and insightful comments, which have helped us to improve the quality of our manuscript. A point-by-point response to the reviewer-1 comments is appended below for your convenience.

Reviewer 2 Report
The issue of regeneration has not been sufficiently worked out, but the authors themselves write about this, and it is not clear why NaOH was used for this. It was also not clear to me how much hydrochloric acid was eventually added during the preparation of the adsorbent. The authors did not indicate how the “separation factor ‘RL’” is calculated and did not give a link to the equation for its calculation, and if the readers are not quite in the subject, then what kind of factor they do not understand. One could also estimate the specific surface area and porosity. So the paper can be published after revision.Author Response
We appreciate the efforts of the reviewers for their detailed and insightful comments, which have helped us to improve the quality of our manuscript. A point-by-point response to the reviewer-2 comments is appended below for your convenience.

Reviewer 3 Report
Comments to the Author Manuscript Catalysts-1584262
The manuscript entitled ‘Bio-stimulated adsorption of Cr(VI) from aqueous solution by Groundnut Shell Activated Carbon@Al embedded material’ by Rao et al focuses on the synthesis of bioadsorbent aluminum metal blended with groundnut shell activated carbon material (Al-GNSC) and it practical application for Cr(VI) adsorption from waste aqueous solutions. Very interesting and well-carried study. The experimental approach is sound. The manuscript is well-organized, all the conclusions are supported by the presented data.
The paper mainly presents two key contributions:
- Synthesis of bioadsorbent and evaluation of its structure using SEM and FT-IR spectroscopy.
- Examination of prepared adsorbents for Cr(VI) ions removal.
There are some points which must be edited or clarified by providing additional information or comments:
- The authors should write the complete terms of all abbreviations (including the instruments) before the first use in the abstract and main manuscript i.e. FT-IR and SEM in abstract section et al.
- The authors should clearly explain the innovation and importance of their work on the introduction of the manuscript. They should justify the value of the work and compare their work with previously similar published papers.
- Fig. 1 - for a more effective visual comparison, authors recommended to provide SEM images of the same scale. In such form is rather difficult to make adequate comparison.
- EDX analysis (Fig 3) - a very conditional (not specific) type of analysis of the chemical composition of the surface. Often, spectra with different atomic abundances of elements can be obtained even from the same sample. First of all, authors have to attach a EDX mapping images before/after sorption of chromium ions. And the authors are strongly recommended to add XPS spectra to this section of revised manuscript. The XPS method is much more sensitive and more accurately determines changes in the chemical composition of samples.
- Why authors did not use XRD technique for sample characterization?
- The adsorption capacities of Al-GNSC adsorbents at different contact times have been provided. Which kinetics are right? Please add missing information about appropriate kinetic model in revised manuscript.
- It is rather difficult to make an adequate comparison of certain properties (catalysts or sorbents) with the already available results, since the concentration of the pollutant and the mass of the loaded sorbent vary in each experiment. Therefore, the authors are recommended to add the missing information (i.e. conditions for testing sorbents of Cr(VI) ions) to Table 3.
- In order to confirm proposed mechanism of Cr(VI) adsorption (illustrated on the fig 6) Authors should provide data on adsorption capacity of pristine groundnut shell activated carbon (not modified with Al).
- The conclusion section should be elaborated and improved. The author should bring specific conclusions in accordance with obtained results.
Our decision on this manuscript – Major revision. After making all required minor changes in article it could be recommended for publication.
Author Response
We appreciate the efforts of the reviewers for their detailed and insightful comments, which have helped us to improve the quality of our manuscript. A point-by-point response to the reviewer-3 comments is appended below for your convenience.

Round 2
Reviewer 1 Report
The paper now can be accpeted for publication.
Reviewer 3 Report
Authors respond for all my queries and in present for this manuscript could be recommended for publication